# Evaluation of Disease Causality of Rare *Ixodes ricinus*-Borne Infections in Europe

**DOI:** 10.3390/pathogens9020150

**Published:** 2020-02-24

**Authors:** Tal Azagi, Dieuwertje Hoornstra, Kristin Kremer, Joppe W. R. Hovius, Hein Sprong

**Affiliations:** 1Centre for Infectious Diseases Research, National Institute for Public Health and the Environment, P.O. Box 1, Bilthoven 3720 BA, The Netherlands; kristin.kremer@rivm.nl (K.K.); hein.sprong@rivm.nl (H.S.); 2Center for Experimental and Molecular Medicine, Amsterdam University Medical Centers Location Academic Medical Center, Amsterdam 1105 AZ, The Netherlands; d.hoornstra@amsterdamumc.nl (D.H.); lyme@amsterdamumc.nl (J.W.R.H.)

**Keywords:** human granulocytic anaplasmosis, neoehrlichiosis, *Borrelia miyamotoi* disease, spotted fever rickettsiosis, babesiosis, tick-borne diseases, *Ixodes ricinus*

## Abstract

In Europe, *Ixodes ricinus* ticks transmit pathogens such as *Borrelia burgdorferi* sensu lato and tick-borne encephalitis virus (TBEV). In addition, there is evidence for transmission to humans from *I. ricinus* of *Anaplasma phagocytophilum, Babesia divergens*, *Babesia microti, Babesia venatorum*, *Borrelia miyamotoi*, *Neoehrlichia mikurensis*, *Rickettsia helvetica* and *Rickettsia*
*monacensis*. However, whether infection with these potential tick-borne pathogens results in human disease has not been fully demonstrated for all of these tick-borne microorganisms. To evaluate the available evidence for a causative relation between infection and disease, the current study analyses European case reports published from 2008 to 2018, supplemented with information derived from epidemiological and experimental studies. The evidence for human disease causality in Europe found in this review appeared to be strongest for *A*. *phagocytophilum* and *B. divergens*. Nonetheless, some knowledge gaps still exist. Importantly, comprehensive evidence for pathogenicity is lacking for the remaining tick-borne microorganisms. Such evidence could be gathered best through prospective studies, for example, studies enrolling patients with a fever after a tick bite, the development of specific new serological tools, isolation of these microorganisms from ticks and patients and propagation in vitro, and through experimental studies.

## 1. Introduction

Lyme borreliosis and tick-borne encephalitis (TBE) are caused by infections with *Borrelia burgdorferi* sensu lato and tick-borne encephalitis virus (TBEV), respectively. These well-established tick-borne diseases are frequently reported in Europe, and several studies have indicated a rise in the incidence of these diseases over the last decades [1,2,3]. In Europe, both pathogens are transmitted to humans by the bite of *Ixodes ricinus*. During such a bite, humans might also be exposed and sometimes infected with other microorganisms transmitted by *I. ricinus* [4]. These potential tick-borne pathogens (TBPs) include *Anaplasma phagocytophilum, Babesia divergens*, *Babesia microti, Babesia venatorum*, *Borrelia miyamotoi*, *Neoehrlichia mikurensis*, *Rickettsia helvetica,* and *Rickettsia monacensis,* which have all been associated with human disease in Europe in case reports [5]. Although the number of case reports describing European human infections and disease associated with these agents is accumulating, the evidence for causality of infection with these microorganisms and human disease has not been fully demonstrated. Whenever a tick-borne pathogen is transmitted to a human host, the disease will not always ensue [6], because detrimental health effects are but one outcome of the interaction between a host and a microorganism [7]. Thus, the term pathogen points to a microorganism having the potential to cause disease following transmission [8].

Some of the *I. ricinus-*borne microorganisms are closely or more distantly related to species that are established disease agents in other tick species or continents. For example, *R. helvetica*, which is frequently encountered in *I. ricinus* throughout Europe, is related to other Spotted Fever Group *Rickettsia*, whose pathogenicity has been undisputedly shown. These include *Rickettsia rickettsii*, the causative agent of Rocky Mountain spotted fever (RMSF), a disease with a typical clinical picture, including a characteristic macular rash [9,10]. Epidemiological studies associate RMSF to *Dermacentor* ticks. Indeed, *R. rickettsii* could be successfully cultured from ticks, but also from patients [11,12]. Furthermore, infected ticks were able to transmit *R. rickettsii* when experimentally fed on animals [13,14], and *R. rickettsii* was found in endothelial cells of vascular lesions in acute cases [12,15]. In addition, Mediterranean spotted fever (MSF) is caused by *Rickettsia conorii* and also presents itself with a typical clinical picture, namely an eschar at the site of the tick-bite [12,16]. *Rhipicephalus sanguineus* ticks were identified as the main vector of *R. conorii,* which could be isolated and cultured both from patients and ticks [17,18,19]. Later, the ability of *R. conorii* to cause endothelial damage was proven in vivo [17,20,21]. In contrast, although *R. helvetica* has been isolated and cultivated from *I. ricinus* and a human patient [22], a typical clinical picture and consistent pathology of patients with a *R. helvetica* infection are lacking, as well as epidemiological evidence of association with disease [23,24,25].

*A. phagocytophilum* was established as the etiologic agent of Human Granulocytic Anaplasmosis (HGA) in the United States of America (USA) [26,27,28]. Patients lack characteristic symptoms, but develop acute fever, malaise, headache, myalgia, and/or arthralgia, often with a recent history of a tick-bite. HGA was initially misdiagnosed as Human Monocytotropic Ehrlichiosis (HME), a disease caused by *Ehrlichia chaffeensis* and transmitted by *Amblyomma americanum* [29,30]. *A. phagocytophilum* has successfully been cultivated from patient blood and *Ixodes scapularis* ticks [29,31,32] and transmission has been demonstrated in experimental settings [27,33]. Furthermore, *A. phagocytophilum* infects granulocytes, inducing leukocytopenia, often accompanied, in humans as well as in experimental animal models, by thrombocytopenia and elevated liver enzymes, e.g., Alanine aminotransferase (ALT) or Aspartate transaminase (AST) [27]. Epidemiological studies in the USA, where HGA is notifiable, show a steadily rising incidence [34]. In contrast to the USA, HGA is not notifiable in European countries, and the number of described cases is relatively low. The differences in the number of identified cases of HGA in the USA and Europe could be caused by limited awareness, but also by differences in vector, host susceptibility, and importantly, the pathogenicity of circulating *A. phagocytophilum* strains.

As long as the ability to cause disease by the European genospecies of potential TBPs has not been firmly established, it is difficult to assess to what extent public health professionals should increase awareness amongst the public and medical professionals and implement routine diagnostic modalities for these microorganisms. However, these public health actions are important prerequisites to be able to adequately assess the severity and incidence of *I. ricinus-*borne infectious diseases other than Lyme Borreliosis and TBE in Europe [35]. 

The nature of these *I. ricinus*-borne agents makes it difficult, if not impossible, to meet Koch’s postulates, as has previously been discussed [35,36,37]. In order to evaluate their involvement in human disease, several pitfalls must be overcome. Firstly, the clinical symptoms, as well as observed abnormalities in blood, attributed to infections with tick-borne microorganisms other than *B. burgdorferi* s.l. and TBEV, are non-characteristic and relatively mild in immunocompetent individuals. Without knowledge of a tick-bite, such a clinical picture could be attributed to any common infectious disease agent, e.g., influenza virus. Secondly, the isolation and propagation of microorganisms from patients is an important prerequisite for the establishment of their pathogenicity. Unfortunately, most of these tick-borne microorganisms are difficult to cultivate, even in dedicated laboratories [38]. For example, the cultivation and in vitro propagation of *B. miyamotoi* [39] and the cultivation of *N. mikurensis* has only been described in the literature recently [40,41]. Although serological or PCR-based methods have been shown to be useful as diagnostic tools for infectious diseases, they provide only limited evidence for (active) infection [42,43,44]. Thirdly, *I. ricinus* is frequently infected with more than one microorganism, enabling human exposure and possibly infection with multiple microorganisms, referred to as co-infections, after a singular or consecutive tick-bite. This could confuse the identification of the causative agent of the disease. For example, erythema migrans (EM), a typical manifestation of Lyme borreliosis, has been reported in patients in which infection with *N. mikurensis* or *B. miyamotoi* was detected by molecular methods [45,46]. This does not mean that these microorganisms are able to cause EM; it is much more likely that the EM is caused by an infection with *B. burgdorferi* s.l. and that the other microorganisms were present in the blood, either giving rise to other symptoms or indolent. Furthermore, the detection of *R. helvetica* and *R. monacensis* DNA and antibodies in patients with or suspected to have Lyme neuroborreliosis is a puzzling phenomenon and it remains to be elucidated whether coinfection with these spotted fever Rickettsiae has any clinical implications at all [24,25]. Fourthly, a final level of complexity is that some of these other *I. ricinus*-borne microorganisms might be able to cause disease only under specific conditions, i.e., when a patient is immunocompromised or has a co-infection. 

Therefore, the aim of this literature study was to evaluate the published European case reports on tick-borne diseases—other than Lyme Borreliosis or TBE—in order to identify which clinical symptoms are associated with probable infections. This information, as well as data from epidemiological and experimental studies, was used to assess whether there is evidence for a causative relation between infection with these tick-borne microorganisms and human disease in Europe. The literature search was limited to the decade between 01-01-2008 and 01-03-2018, in order to assess the current situation. Our study focused only on microorganisms found in the European *I. ricinus* ticks and molecularly proven to be transmitted to their human host after a tick-bite, and not on microorganisms found in *I. scapularis* or *I. persulcatus* [6]. 

## 2. Results

### 2.1. Literature Search

Our literature search resulted in 3366 unique titles, out of which 3303 did not meet our search criteria based on the abstracts (Appendix A), thus yielding 63 papers. In addition to the literature search, we manually included four additional case studies as part of a structural search. These, in total, 67 remaining papers were screened by reading the full text and 60 published case studies were found relevant for this study (Figure 1). These 60 European case studies, from between January 2008 and March 2018, described 77 individual cases. *A. phagocytophilum* case reports were most common, accounting for 23 of the papers and 32 cases. This was followed by 20 *Babesia* case reports describing 21 cases, of which 11 were associated with *B. divergens*, while seven and three were suspected of *B. microti* or *B. venatorum* infection, respectively. A total of 8 publications described 14 *N. mikurensis* cases, while 6 publications described 3 and 4 cases suspected of *R. helvetica* or *R. monacensis* infection, respectively. Finally, three *B. miyamotoi* cases were described in three publications. Infection with the various *I. ricinus* borne microorganisms was confirmed by molecular biology methods such as PCR, serology, and microscopy (Table 1).

### 2.2. Extracted Data

#### 2.2.1. Anaplasma Phagocytophilum

##### Clinical Presentation

The most common symptoms reported in the 32 European cases of anaplasmosis analyzed [47,48,49,50,51,52,53,54,55,56,57,58,59,60,61,62,63,64,65,66,67,68,69] were fever, headache, malaise, and myalgia (41%–91%). Arthralgia and erythematous skin manifestations such as a maculopapular rash were less common (25% and 22%, respectively), and fewer cases presented with lymphadenopathy, chills, rigors, abdominal complaints, anemia, splenomegaly, and darkened urine (3%–19%) (Table 2). General laboratory findings included elevated C-reactive protein (CRP) values, leukocytopenia, thrombocytopenia, elevated liver enzymes and lactate dehydrogenase (LDH) values (Appendix A). One patient was immunocompromised, and hospitalization was required in 88% of cases. Most patients (94%) received antibiotic treatment mainly in the form of doxycycline and subsequently recovered. In two cases, the outcome was fatal.

##### Exposure to Ticks

Eleven (34%) patients recalled a tick-bite, and 12 (38%) of the cases reported exposure to ticks.

##### Culture from Ticks or Patient Materials

Although *A. phagocytophilum* was not cultured in any of the case reports analyzed in this study, culture in a HL-60 promyelocytic cell line from a European patient has been successfully performed in the past in the Czech Republic [70].

##### Pathology

Both the European and American variants of *A. phagocytophilum* have been shown to infect human and animal host reservoir neutrophils, hence the name Human Granulocytic Anaplasmosis (HGA) [42,71,72]. The disruptive effect on infected neutrophils may lead to detrimental effects on the host’s immune response and poor regulation of inflammatory processes. This is believed to give rise to secondary pathological findings such as tissue damage in the spleen, liver, and kidneys [71]. In farm animals, such as sheep and cattle, the severity of the symptoms was affected by factors such as the host immune status or pre-existing infections [73]. In the cases evaluated in this study, the primary damage to neutrophils and secondary organ damage can be deducted from the reported laboratory test results such as leukocytopenia and elevated liver enzymes, respectively (Table 2). Moreover, bacteria were visualized in 25% of cases in Giemsa stained blood or bone marrow smears.

##### Transmission Experiments

Transmission of a Dutch strain of *A. phagocytophilum* from infected *I. ricinus* to naïve dogs has been shown, as well as the acquisition of the bacteria by *I. ricinus* in vitro [74].Moreover, a longitudinal study on sheep-to-tick transmission efficiency under natural conditions found molecular evidence of *I. ricinus* infected with *A. phagocytophilum* after feeding on their sheep host [75]. Moreover, the bacterium has been detected in *I. ricinus* ticks as well as animal reservoirs in multiple European countries [76,77,78,79]. Moreover, human tick-bite studies have shown an association between a tick-bite by *I. ricinus* ticks and the presence of *A. phagocytophilum* in patient blood. For example, a study in which 626 blood samples of Dutch patients who reported a tick-bite, were analyzed for the presence of tick-borne microorganisms found five samples positive for *A. phagocytophilum* by molecular screening [6]. Another study in Poland, in which whole blood, serum, and cerebrospinal fluid (CSF) samples from 118 symptomatic patients who reported a tick-bite were screened for tick-borne microorganisms, reported molecular detection of *A. phagocytophilum* in 14 of the samples [80]. These data support the assumption that this microorganism can be transmitted from *I. ricinus* to both animals and humans.

#### 2.2.2. *Babesia* Species

##### Clinical Presentation

In the seven cases where *B. microti* was reported as the probable cause of disease [81,82,83,84,85,86,87], fever was reported for all patients, anemia in all except one, and also malaise, chills, headache, and abdominal complaints were relatively common (ranging from 57% to 43%). Skin manifestations were reported only in one case (Table 2). General laboratory findings included elevated CRP, thrombocytopenia, elevated liver enzymes, bilirubin, and LDH (Appendix A). Hospitalization was required in 86% of the cases. No fatal cases were described between 2008 and 2018. All patients received treatment with a combination of antibiotics and antiparasitic medication such as azithromycin-atovaquone or proguanil and quinine. It is important to note that infection with *B. microti* was acquired outside of Europe, namely North America and Uruguay, and solely diagnosed and/or treated in Europe.

In ten additional publications [88,89,90,91,92,93,94,95,96,97], 11 babesiosis cases were reported where infection with *B. divergens* was the probable cause of disease, and fever was the most common symptom (82%). Anemia, malaise, jaundice, abdominal complaints and darkened urine were all relatively common (ranging from 45% to 27%), while headache, myalgia and arthralgia were reported for only two patients (Table 2). General laboratory findings included elevated CRP values, leukocytopenia, thrombocytopenia, and elevated liver enzymes, bilirubin, LDH and kidney function (Table 1 and Appendix A). Hospitalization was required in 91% of the 11 cases, 64% were immunocompromised, out of which 54% were splenectomized. Treatment was administered to 91% of patients. In one case, doxycycline was provided while most followed the guideline advised dual therapy with an antibiotic and antiparasitic medication. Of the nine cases that received combination-therapy, three were fatal, two of which were immunocompromised. Additionally, an immunocompromised patient without treatment died.

Finally, three cases were associated with *B. venatorum* infection [98,99,100]. Symptoms included anemia (100%), fever, darkened urine (67%), rigors and myalgia (33%). General laboratory findings included elevated CRP values, to a lesser extent thrombocytopenia, and elevated bilirubin, LDH, and kidney function. (Appendix A). One case required hospitalization, and in all cases, treatment with quinine and clindamycin or azithromycin and atovaquone was followed by recovery. All three patients were immunocompromised and had undergone splenectomy. 

##### Exposure to Ticks

In *B. microti* suspected cases, patients recalled a tick-bite in 14% of the cases, and exposure to ticks was reported in the remaining 86% of the cases. For cases where *B. divergens* infection was detected, patients recalled a tick-bite in 45% of cases and exposure to ticks was reported in the other 55% of cases. Exposure to ticks was reported in 33% of the *B. venatorum* suspected cases. 

##### Culture from Ticks or Patient Materials

None of the *Babesia* species were cultured in the case reports analyzed. However, *B. divergens* has been successfully cultured, propagated, and sequenced from samples deriving from a French patient and it has also been cultivated from European cattle [101,102,103]. *B. microti* has been isolated from field voles in Germany, and *B. venatorum* has been cultured from roe deer erythrocytes in France [104,105].

##### Pathology

*Babesia* spp. are intraerythrocytic parasites, and symptoms such as anemia, jaundice, and hemoglobinuria (presenting as darkened urine) are a result of hemolysis [106].

In the cases evaluated for *B. microti, B. divergens,* and *B. venatorum,* intra-erythrocytic parasites could be visualized in Giemsa stained blood smears of 91%, 86%, and 33% of patients, respectively. Parameters linked to hemolysis, such as anemia and elevated bilirubin, were also reported. Thrombocytopenia was common in *B. microti* (86%) and *B. divergens* (73%) cases and less frequent in *B. venatorum* (33%) cases.

##### Transmission Experiments

A European strain of *B. microti* was transmitted in vivo from *I. ricinus* to gerbils and vice versa [107]. *B. divergens* was transmitted to *I. ricinus* ticks through feeding on inoculated gerbils and cattle [108]. Moreover, *B. venatorum* was transmitted to *I. ricinus* ticks fed on infected sheep blood and vice versa [109]. These findings demonstrate that *I. ricinus* is a competent vector for all three *Babesia* species.

#### 2.2.3. Borrelia miyamotoi

##### Clinical Presentation

Three European cases with suspected *B. miyamotoi* disease (BMD) were included in this study [110,111,112], of which two patients were immunocompromised. Symptoms reported included neurological complaints in the two immunocompromised cases due to meningoencephalitis and headache (67%). Fever, abdominal complaints, myalgia, arthralgia, weight loss, and lymphadenopathy were observed in one case (33%). The two immunocompromised patients received antibiotic treatment (ceftriaxone) and recovered. The remaining patient recovered without treatment and showed elevated CRP values, leukocytopenia, and thrombocytopenia.

##### Exposure to Ticks

Patients recalled tick-bites in all three cases.

##### Culture from Ticks or Patient Materials

*B. miyamotoi* was isolated and cultivated in modified Kelly–Pettenkofer medium supplemented with 10% fetal bovine serum from *I. ricinus* egg masses resulting in two Dutch isolates, of which one was propagated and both were sequenced [113].

##### Pathology

As opposed to the hematogenous disease commonly seen in immunocompetent cases, in immunocompromised patients, BMD has been reported to manifest neurologically (e.g., meningoencephalitis) [114]. However, recently (after the study period of this review) two new neurological BMD cases have been diagnosed in Sweden, of which one case was immunocompetent [115]. Neurological involvement is demonstrated in the CSF by pleocytosis, visible spirochetes by dark-field microscopy and by molecular analysis specific for relapsing fever *Borrelia* and *Borrelia miyamotoi* [115]. Pleocytosis and visible spirochetes in the CSF by dark-field microscopy were also reported in the Dutch immunodeficient case included in this study [110,114].

##### Transmission Experiments

A Dutch tick isolate of *B. miyamotoi* has been shown to be infective to *I. ricinus* fed on infected CD1 mice and vice versa (unpublished) [116]. Moreover, field-collected *B. miyamotoi* infected *I. ricinus* larvae have been shown to transmit the spirochete to naive rodents [117]. This suggests *I. ricinus* is a competent vector for *B. miyamotoi* transmission.

#### 2.2.4. Neoehrlichia mikurensis

##### Clinical Presentation

The most prevalent symptom in the 14 cases of neoehrlichiosis included [46,118,119,120,121,122,123] fever (86%), less common were skin manifestations (erythematous rash and EM) and anemia (36% and 29%, respectively), and in fewer cases, symptoms such as malaise, chills, arthralgia, abdominal complaints, edema, myalgia, and weight loss were reported (ranging from 14% to 7%). General laboratory findings included elevated CRP values, leukocytopenia, thrombocytopenia, and elevated liver enzymes (Appendix A). Hospitalization was required for 79% of the patients. All patients received treatment mainly in the form of doxycycline. Two cases received phenoxymethylpenicillin, and one case received piperacillin-tazobactam, ciprofloxacin, and fosfomycin. All cases but the latter recovered. The majority of cases (n = 10; 71%) were immunocompromised, and five of these (50%) had undergone splenectomy.

##### Exposure to Ticks

Five (36%) patients recalled a tick-bite and two (14%) of the cases reported exposure to ticks.

##### Culture from Ticks or Patient Materials

Only recently, *N. mikurensis* was reported to be cultivable in human primary endothelial cells derived from skin microvasculature as well as pulmonary arteries and in the *I. ricinus* IRE/CTVM20 cell line. Cultivation was performed using PCR-positive blood from Swedish patients [41].

##### Pathology

The pathology of neoehrlichiosis is not yet fully understood. As *N. mikurensis* was isolated and propagated in human primary endothelial cells and was visualized in the cytoplasm of circulating endothelial cells of infected patients, it has been suggested that endothelial cells are the infectious target for the bacteria [41]. There is evidence in reported cases for vascular events, such as deep vein thrombosis, thromboembolic events, and transitory ischemic accidents [124]. Moreover, two of the cases in this study presented with arterial aneurysms [119,122].

##### Transmission Experiments

*N. mikurensis* has been widely molecularly detected in *I. ricinus* in multiple European countries, such as Belgium, France, Germany, The Czech Republic, Portugal, Poland, The Netherlands, and Sweden [125,126,127,128]. Moreover, the bacterium has been isolated from xenodiagnostic *I. ricinus* larvae fed on infected rodents, further implicating this tick species serves as a vector for the bacteria [129]. As far as we are aware, no studies have been published on tick transmission in animal models.

#### 2.2.5. Spotted Fever Rickettsiae

##### Clinical Presentation

Three *R. helvetica* case reports were included [22,130,131]. All these cases developed fever and headache. Accompanying photophobia was common (67%), while skin manifestations (such as a macular rash), myalgia and arthralgia, were presented in only one case. Laboratory findings included elevated CRP values, elevated erythrocyte sedimentation rate (ESR), leukocytopenia, thrombocytopenia, and elevated liver enzymes and kidney function (Appendix A). The three cases required hospitalization and recovered following antibiotic treatment with doxycycline.

In the four cases where infection with *R. monacensis* was reported as the probable cause of disease [132,133,134], skin manifestations (such as erythematous rash and EM) were reported. One of the cases, in which the EM was reported, was found to be co-infected with *B. afzelii*. Fever and headaches were common symptoms (75%) and anemia was reported in one patient. Laboratory findings included leukocytopenia, as well as elevated ESR and bilirubin (Appendix A). One patient required hospitalization, but all patients recovered following antibiotic treatment with doxycycline and in one case with azithromycin. 

##### Exposure to Ticks

No tick-bites were reported in the *R. helvetica* cases, however, all patients had been exposed to ticks. Of the *R. monacensis* cases 50% reported a tick-bite and 25% were exposed to ticks.

##### Culture from Ticks or Patient Materials

*R. helvetica* has been cultivated in African green monkey kidney fibroblast cells (VERO) from CSF of two Swedish cases included in this study [22,131], of which one was also infected by Herpes simplex virus. The bacterium has also been successfully cultured and sequenced from questing *I. ricinus* ticks in VERO cells [135,136]. *R. monacensis* was isolated from a patient’s blood in Spain and cultivated in VERO cells [132]. Moreover, isolation of *R. monacensis* from *I. ricinus* ticks in VERO, mouse fibroblast (L-929), and tick (IRE11, DAE100, ISE6) cell lines have been reported [137].

##### Pathology

*R. helvetica* DNA has been detected by using a semi-nested PCR of the 16S rRNA and 17-kDa outer-membrane protein genes, in the pericardium and in a lymph node from the pulmonary hilum in one patient and in a coronary artery and the heart muscle of another, in two Swedish cases of sudden cardiac death postulated to be due to a *R. helvetica* infection. The bacteria were also visualized in macrophages and endothelial cells using Gimenez and May-Grünwald-Giemsa (MGG) staining, wherein their structure was confirmed by transmission electron microscopy [138]. *R. helvetica* has also been visualized in granulomatous endothelial tissue and macrophages from Swedish patients with sarcoidosis, even though an association with the disease was not established [139]. Additionally, *R. helvetica* DNA has been detected in CSF of patients in The Netherlands and Denmark [24,140]. In two Danish cases, *R. helvetica* was cultured from CSF, leukocytopenia was observed, and a macular rash was reported for one of the patients, supporting the findings above. The pathology of *R. monacensis* has not been studied, but the rashes described in the case reports analyzed, suggest endothelial tissue tropism.

##### Transmission Experiments

Both *R. helvetica* and *R. monacensis* have been repeatedly detected by molecular means both from *I. ricinus* ticks and, to a lesser extent, from wildlife in various European countries [137,141,142,143,144,145,146,147,148,149]. As far as we are aware, there are no published transmission experiments in laboratory animals for either species.

## 3. Discussion, Conclusions and Knowledge Gaps

Since Koch’s postulates are not applicable to all potential disease agents, alternatively, the sum of various alternative approaches could provide sufficient evidence to support causality between a microorganism and human disease, in this case, a tick-borne disease. Such has been the succession of events for several established tick-borne pathogens, for example, for *B. burgdorferi* s.l. and TBEV. Case reports can provide extensive qualitative, but not quantitative data, thus showing circumstantial, but insufficient scientific evidence for disease causality. The cases presented are usually exceptional ones, such as immunocompromised patients, and because the studies are retrospective in nature, the reports suffer from reporter bias. As part of the current study, by supplementing the data from case reports on *A. phagocytophilum*, *B. venatorum*, *B. divergens*, *B. microti*, *B. miyamotoi*, *N. mikurensis*, *R. helvetica,* and *R. monacensis* with epidemiological and experimental studies, we have assessed their causal relationship to human disease. Thus, we found that to varying degrees, the existing evidence is incomplete for all the microorganisms mentioned above, but seemed strongest for *A. phagocytophilum* and *B. divergens*. This approach also provided a comprehensive outlook on the current knowledge gaps that need to be filled (Table 3).

*A. phagocytophilum* causes illness in immunocompetent patients in Europe, and the case reports amounted to half of all papers included in this review. The clinical picture, laboratory test results, and low fatality rate were reminiscent of what has been described for HGA in the USA [26,43]. A rash was reported in a minority of patients, which is not considered to be a specific sign of HGA and may point to co-infections with other tick-borne bacteria, such as *Rickettsia* or *Borrelia* spp. [134,153,154,155], as was confirmed for one of these cases [56]. In Europe *A. phagocytophilum* has been cultivated from a patient’s blood and its tissue tropism, namely neutrophils, has been demonstrated [42,70]. *A. phagocytophilum* has not been isolated from *I. ricinus*. However, there is ample evidence of its association with *I. ricinus* ticks [76,77] and evidence that this tick species can be infected by natural reservoirs [75].

In accordance with published data [156,157], most cases reporting *Babesia* infection in Europe were attributed to *B. divergens*, and, while disease has been associated with immunocompromised and splenectomized patients [157], almost half of the cases were immunocompetent. Four cases were fatal, and three of these were immunocompromised and one was immunocompetent and had a co-existing Lyme borreliosis infection [90]. Co-infections with *Babesia* are a known possibility in Lyme borreliosis patients and should be considered in cases from endemic regions presenting with high fever, anemia, thrombocytopenia, leukocytopenia, or other hemolysis signs, or in cases who do not respond to treatment [158]. Co-infection of *B. microti* and *B. burgdorferi* s.l. has been found to increase both the severity and duration of illness in North America [153,159,160], such correlations have not been shown for co-infection with *B. divergens* and *B. burgdorferi* s.l. Our analyses also suggested that *B. microti* is able to cause disease in Europe, but mainly as an imported agent presenting as a non-fatal illness in immunocompetent patients. Although it is present in *I. ricinus* and European wildlife and has been detected in human samples from Germany, Poland, and Switzerland [157], the number and severity of *B. microti* cases, be it from lack of awareness or genetic difference, is low when compared to North America [157]. In the USA, 2358 cases were reported in 2017 and fatality rates of up to 6%–9% have been reported in immunocompetent patients [156,161]. The first European originated case, probably acquired by a blood transfusion, was published in 2007 [162], however, the protozoa have not been cultivated from European patients or *I. ricinus.* Cultivation of European strains would facilitate genomic analyses in the search for differences in pathogenicity when compared to North American strains. The few *B. venatorum* cases reported were relatively mild and affected immunocompromised patients. *Babesia venatorum* can be transmitted by *I. ricinus* in laboratory settings [109]. However, cultivation from ticks and patients remains to be performed and would also assist the development of specific serological tests for diagnostic purposes and future population studies.

*Borrelia miyamotoi* is an emerging tick-borne microorganism whose potential to cause disease has only recently been disclosed [114]. Since the first case reports in 2011 in Russia, over 450 cases have been published, mainly from Asia and North America, with a total of 5 cases from Europe. *B. miyamotoi* is able to cause an acute febrile disease with flu-like symptoms. Skin lesions, such as EM, have been reported in *B. miyamotoi* suspected cases, but are presumed to be related to a co-infection with the etiologic agent of Lyme disease, *B. burgdorferi* s.l. In immunocompromised cases, of which most originate from Europe, the disease seems to develop neurologically, however, a meningoencephalitis in an immunocompetent case has been reported. Nevertheless, more European cases need to be evaluated to assess the full extent of the clinical symptoms and effectiveness of the available diagnostic modalities. *B. miyamotoi* has not been cultivated from an individual infected by an *I. ricinus* tick-bite. However, it has been cultivated from 12 Russian patients, some from European Russia, of which 6 were sequenced and published [163], with *I. persulcatus* being the most likely vector. Cultivation of more *B. miyamotoi* isolates from *I. ricinus* and European human cases would enable comparative genomic analyses between isolates from Europe, North America, and Asia [164]. This could be an enormous impetus for basic, molecular, and epidemiological research and could lead to the development of specific diagnostic tools for this emerging potential TBP.

Another emerging potential TBP is *N. mikurensis* [124]. The findings of this study support the growing evidence of *N. mikurensis* as a disease agent affecting the immunocompromised population, and show it is also found in immunocompetent patients. EM was reported in two cases where co-infection with *A. phagocytophilum* in one and *B. burgdorferi* s.l. in the other was present [46]. A Norwegian study, in which EM patients were screened by PCR for the presence of *N. mikurensis,* reported seven positive samples, of which only three were positive for *B. burgdorferi* s.l. by serology [165]. However, this could be explained by the low diagnostic accuracy of serology in early Lyme borreliosis [166]. Three other cases analyzed in this study, of immunocompromised patients presenting with a rash, were diagnosed by a positive *N. mikurensis* PCR, and in two cases by transmission electron micrographs. However, no serology was performed in order to rule out a co-infection that could have caused the rash [120,121]. Although *Ehrlichia* species are known to infect granulocytes or mononuclear phagocytes [167], the bacterium has been cultivated from patient’s blood in a tick cell line and in human primary endothelial cells [41]. Routine cultivation, as well as cultivation from ticks would enable development of the serological assays, which are currently lacking from the routine practice.

Consistent with it being a spotted fever group *Rickettsia*, three *R. monacensis* cases presented with a rash, and in one case in which an EM was present, a co-infection with *B. afzelii* was detected [134]. However, no studies have been performed to elucidate the specific pathology of this microorganism. The clinical picture reported for *R. helvetica* in this study included headaches and photophobia, which is suggestive of neurological involvement. One of the cases presented with subacute meningitis [22] and although the diagnosis of neuroborreliosis was rejected based on serology, the test was performed in an early disease stage when test sensitivity infamously falls short. Another case presented with meningoencephalitis [131] and was positive for Herpes Simplex Virus 2, which could account for the clinical picture. In the three case reports included in the study *R. helvetica* was either isolated and cultivated or isolated, accompanied by seroconversion indicating that this microorganism can cause infection in humans. However, proof found of a Rickettsia infection does not imply or translate to symptomatic disease. Therefore the evidence should not be mistaken for disease causation, which has not been demonstrated [22,131]. Interestingly, *R. helvetica* has been detected as a co-infection with Lyme neuroborreliosis both in patient blood and CSF in a Dutch study [24]. *R. helvetica* has also been found as a co-infection in Danish Lyme borreliosis and neuroborreliosis patients [140], however the involvement of this bacterium with neurological symptoms has not been studied. Although both *R. helvetica* and *R. monacensis* have been isolated from patients and *I. ricinus* ticks, only few case reports have been published on cases caused by these species and no transmission experiments have been performed to date, in contrast to, for example, *R. conorii,* an established human pathogen in Europe.

In all case reports analyzed in this study, diagnostic tools such as microscopy, serology, and molecular analysis were required in order to reach a diagnosis. In some cases, these tools were employed retrospectively, as they were not accessible in routine practice. Moreover, the cultivation of microorganisms such as *A*. *phagocytophilum*, *N. mikurensis,* and *Babesia* spp. was not attempted as their growth requires dedicated settings, is technically demanding, and time-consuming. Even in experimental settings, cultivation of these microorganisms can be arduous. To date, cultivation from *I. ricinus* has been reported only for *B. miyamotoi*, *R. helvetica,* and *R. monacensis* [113,135,137] and cultivation from *I. ricinus* acquired cases has not been performed for *B. venatorum, B. microti,* and *B. miyamotoi.*

The additional evidence needed in order to conclusively appraise disease causality and elucidate the impact of *I. ricinus* transmitted infections in Europe for these tick-borne microorganisms can be achieved by the fulfillment of several steps. Firstly, to enhance our understanding of the tissue tropism and specific pathology of microorganisms, such as *R. helvetica*, *R. monacensis,* and *N. mikurensis*, experimental transmission experiments using laboratory animals should be performed. Secondly, new and specific serological tools should be developed for the diagnosis of *I. ricinus*-borne microorganisms. This could be facilitated by cultivation of the tick-borne microorganisms from patients or ticks, followed by next-generation sequencing and bioinformatic analyses, for the discovery of new potential antigens. Thirdly, prospective studies need to be performed including patients with febrile illness after a tick-bite, the common denominator in the clinical manifestation of the different tick-borne microorganisms.

## 4. Materials and Methods

### 4.1. Literature Search 

The literature search was performed using the Pubmed and Embase databases (Appendix A). In addition, a structural search was performed in order to find additional cases and information. 

### 4.2. Selection Criteria

The search focused on suspected disease cases with Anaplasma phagocytophilum, Babesia divergens, Babesia microti, Babesia venatorum, Borrelia miyamotoi, Neoehrlichia mikurensis, Rickettsia helvetica, and Rickettsia monacensis. Case reports and series of patients diagnosed in Europe, published in the period between 01-01-2008 and 01-03-2018, were included. The process was performed independently by TA and DH, and when disagreements occurred, the study was re-evaluated by both TA and DH under the supervision of HS, KK, and JH. Publications written in English, Dutch, and Spanish were included. 

### 4.3. Data Extraction

Parameters extracted for all case reports/series were: Sex, age, immune status, underlying health conditions, pre-existing treatments, history of a tick-bite or exposure to ticks, clinical symptoms, general laboratory findings, hospitalization, treatment, disease outcome, diagnostic evaluation, excluded causes of disease and co-infections (Appendix A). Within clinical symptoms, “fever” was attributed to a case whenever the measured temperature exceeded ≥ 37.5 ℃ (99,5 ^o^F) [168] or when the authors reported fever as a symptom. History of a tick-bite was recorded whenever a patient recalled a tick bite on presentation. Exposure to ticks was recorded whenever a patient worked in a high-risk occupation, e.g., forest ranger, had engaged in activities considered to be high risk for a tick bite such as hiking, or had been exposed to animals.

### 4.4. Data Analysis

For each tick-borne microorganism, cases were evaluated for evidence of a causal relationship between the microorganism and the disease described by establishing the nature and frequency of five criteria: The clinical presentation, association to tick-bite(s), culture from ticks or patient materials, transmission experiments, and the ability to induce specific pathology. In addition, for the analyses on cultivation from ticks or patients, transmission experiments and pathology, information was added by selectively searching the literature.

## Figures and Tables

**Figure 1 pathogens-09-00150-f001:**
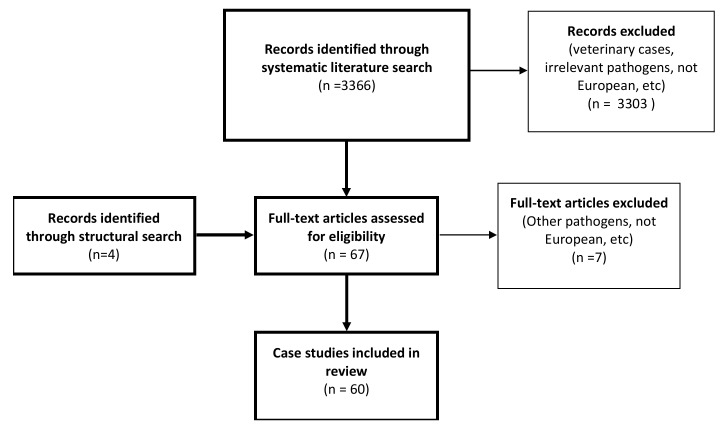
Results of the systematic and structural literature search for case reports of *I. ricinus*-borne microorganisms in the period of January 2008-March 2018. The literature search was performed by two independent authors and the figure shows the selection process involved.

**Table 1 pathogens-09-00150-t001:** Summary of the molecular biology methods performed to confirm infection with *I. ricinus*-borne microorganisms in the case reports included in this study.

Microorganism	Number of Cases	PCR (*Sequenced)	Serology	Blood Smear	Bone Marrow Smear	Other Microscopy	Cultivation
*A. phagocytophilum*	32	23 (*17)	26	7	2	-	-
*B. microti*	7	7 (*5)	2	6	-	-	-
*B. divergens*	11	7 (*7)	5	10	-	-	-
*B. venatorum/*EU1	3	3 (*3)	1	2	-	-	-
*B. miyamotoi*	3	2 (*2)	1	-	-	2	-
*N. mikurensis*	14	14 (*13)	-	-	-	2	-
*R. helvetica*	3	3 (*3)	2	-	-	-	2
*R. monacensis*	4	4 (*4)	3	-	-	-	1

*:Sequenced PCR results.

**Table 2 pathogens-09-00150-t002:** Overview of European cases of *I. ricinus*-borne diseases between 1 January 2008 and 1 March 2018, based on the data of case reports (*) and of other studies (**) retrieved in the systematic literature search of this study. The table describes the number of cases published, clinical manifestations from case reports, exposure to ticks in case reports, cultivation, and propagation from patients or tick materials, evidence from case reports reflecting the pathology in animal models or in vitro observations and transmission experiments in laboratory or wild animals.

Microorganism	Number of Cases *	Clinical Manifestations *	Exposure to Ticks *	Culture from Ticks or Patients **	Pathology *	Transmission Experiments **
*A. phagocytophilum*	32 [46,47,48,49,50,51,52,53,54,55,56,57,58,59,60,61,62,63,64,65,66,67,68]	Fever (88%), headache (56%), malaise (41%), myalgia (41%), arthralgia (25%), erythematous skin manifestations such as morbilliform and maculopapular rashes (22%), lymphadenopathy (19%), abdominal complaints (16%),anemia (13%), chills (13%), splenomegaly (9%), rigors (6%), dark urine (3%)	History of tick-bite (34%), exposure to ticks (38%)	[70] (Cultivated from patient blood)	Leucocytopenia (50%), positive blood/bone marrow smear (25%).	[74](infection of *I. ricinus* from sheep and to naïve dogs), [75] (infection of *I. ricinus* from sheep under natural conditions)
*B. microti*	7 [79,80,81,82,83,84,85]	Fever (100%), anemia (86%), malaise (57%), chills (57%), headache (43%), abdominal complaints (43%), skin manifestation (erythematous rash) (14%)	History of tick-bite (14%), exposure to ticks outside Europe (86%)	None	anemia (86%), thrombocytopenia (86%), elevated liver enzymes (86%), elevated kidney function (14%),	[107] (infection of *I. ricinus* from and to gerbils)
*B. divergens*	11 [86,87,88,89,90,91,92,93,94,95]	Fever (82%), anemia (45%), malaise (36%), abdominal complaints (36%), jaundice (36%), dark urine (27%), headache (18%), myalgia (18%), arthralgia (18%)	History of tick-bite (45%), exposure/risk (55%)	[102] (Cultivated from patient blood)	Thrombocytopenia (73%), elevated kidney values (55%), elevated liver enzymes (45%), anemia (45%)	[108] (infection of *I. ricinus* to cattle and gerbils)
*B. venatorum/EU1*	3 [96,97,98]	Anemia (100%), fever (67%), dark urine (67%), myalgia (33%), rigors (33%)	Exposure/risk (33%)	None	Anemia (100%), elevated kidney values (100%) and thrombocytopenia (33%)	[109] (infection of *I. ricinus* from and to sheep blood)
*B. miyamotoi*	3 [107,108,109]	Neurological complaints (67%), headache (67%), abdominal complaints (33%), myalgia (33%), arthalgia (33%), skin manifestations (33%), lymphadenopathy (33%), weight loss (33%)	History of tick-bite (100%)	[113] (Cultivated from *I. ricinus*)	Unknown	[117] (infection from wild *I. ricinus* to naive mice), [116] (infection of *I. ricinus* from and to CD1 mice)
*N. mikurensis*	14 [43,112,113,114,115,116,117]	Fever (86%), skin manifestation (erythematous rash, erysipelas-like rash and EM; 36%), anemia (29%), malaise (14%), arthralgia (14%), chills (14%), abdominal complaints (14%), oedema (14%), weight loss (7%), myalgia (7%)	History of tick-bite (36%), exposure/risk (14%)	[41] ( Cultivated from patient blood)	Unknown	[129] (infection from wild rodents to *I. ricinus*)
*R. helvetica*	3 [19,126,127]	Fever (100%), headache (100%), photophobia (67%), skin manifestation (macular rash) (33%), myalgia (33%), arthralgia (33%)	Exposure/risk (100%)	[22,131] (Cultivated from patient CSF), [135](Cultivated from *I. ricinus*)	Unknown	None
*R. monacensis*	4 [128,129,130],	Skin manifestation (100%) (erythematous rash, maculopapular rash and EM), fever (75%), headache (75%), anemia (25%)	History of tick-bite (50%), exposure/risk (25%)	[132] (Cultivated from patient blood), [137](Cultivated from *I. ricinus*)	Unknown	None

**Table 3 pathogens-09-00150-t003:** Summary of existing evidence for *I. ricinus* borne microorganisms disease causality in Europe and current knowledge gaps based on the systematic literature search of this study and case reports between 1 January 2008 and 1 March 2018. The table describes the number of European derived cases, cultivation, and propagation from patients or tick materials, evidence from case reports reflecting the pathology in animal models or in vitro observations and transmission experiments in laboratory or wild animals. In addition, shown for comparison *B. burgdorferi* s.l. and TBEV, both established TBPs [150,151,152].

Tick-Borne Microorganism	European Derived Cases	Cultivated from		Pathology	Transmission Experiments	
		Ticks	Patients			
*B. burgdorferi* s.l.	+	+	+	+	+	
TBE virus	+	+	+	+	+	
*A. phagocytophilum*	+	-	+	+	+/&	
*B. divergens*	+	-	+	+	+	
*B. microti*	-	-	-	+	+	
*B. venatorum*	+	-	-	+	+	
*B. miyamotoi*	+	+	-	-	+/#	
*N. mikurensis*	+	-	+	-	-	
*R. helvetica*	+	+	+	-	-	
*R. monacensis*	+	+	+	-	-	

+: Published evidence, -: No evidence, +/&: Transmission experiments with wild animals/ticks and laboratory animals, +/#: Published and unpublished evidence.

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
