# Peer review of "Evaluation of Disease Causality of Rare *Ixodes ricinus*-Borne Infections in Europe"

_pathogens, 2020, doi:10.3390/pathogens9020150_

Round 1
Reviewer 1 Report
There are several Ixodes ricinus tick – associated microorganisms, which are for one reason or another considered potentially pathogenic for human. The review/metanalysis of Azagi et al. seeks for evidence in scientific publications in order to assess the probability of these potential pathogens to cause human disease, which is definitely a highly valid and important question to be addressed. Furthermore, the study identifies several important knowledge gaps to be addressed by future research.
The manuscript is in general well structured and comprehensive. The results of the study are well arranged and presented and subsequently properly discussed addressing possible weak points and limitations as well as drawing overall conclusions based on current level of knowledge.
I have just one general comment to the results section. It is an impressive amount of data that was extracted and summarized by the authors. From the description of the search criteria it seems, that all the cases were confirmed by molecular biology methods. I am convinced this is another important piece of information from the clinical and also research point of view: sample type, timing of sampling, extraction and detection method used. Would it be possible to include this section to each of the pathogens?
There are several minor specific remarks listed bellow:
Introduction
Line 76: Anaplasma strains – Anaplasma phagocytphilum strains?
Line 87 (and elsewhere): B. burgdorferi sl => B. burgdorferi s. l.
Line 105: Rickettsia’s => Rickettsia?
Results
Figure 1 – low quality (at least in the pdf generated for peer review)
Line 138: an erythematous skin manifestations => erythematous skin manifestations
Line 148: “…and exposure to ticks was reported in an additional 38% of cases“ => „…and exposure to ticks was reported in additional 38% of cases“ = so in total exposure to ticks was reported in 34 + 38 = 72% of cases?
Line 156: „infect human and host reservoir neutrophils“ => infect human and reservoir host or animal host neutrophils?
Line 161: existing infection? = . pre-existing infection, co-infection?
Line 166: There is a report on transmission of canine strains of A. phagocytophilum to dogs by infected I. ricinus: https://parasitesandvectors.biomedcentral.com/articles/10.1186/s13071-019-3396-9 ; probably an European strain of A. phagocytophilum? + line 374,375
Line 235: Three European cases with B. miyamotoi suspected disease => with suspected B. miyamotoi disease (BMD)
Line 289: N. mikurensis has been widely isolated from I. ricinus => N. mikurensis has been widely detected in I. ricinus?
Line 296: R. helvetica => R. helvetica
Discussion
Line 383: in cases from endemic regions with high fever => in cases from endemic regions and presenting with high fever
Line 410-411: has not been cultivated from an I. ricinus acquired individual => From an individual infected by I. ricinus tick bite
Line 434: to elucidated on the specific pathology => to elucidate the specific pathology
Line 437: serologically discarded => rejected based on serology
Line 438: on an early disease stage => in an early disease stage
Line 442: „This however should not be mistaken with disease 442 causation which has not been demonstrated“ – I believe I understand what the authors try to say, but the sentence could be formulated better to improve understanding.
Line 467: bioinformatical analysis => bioinformatic analyses
Materials and Methods
Data extraction:
Could you define the terms “tick bite” and “exposure to ticks” you use? E.g. the time frame of a tick bite? Does exposure mean a historical tick bite or a stay in tick habitat?
Line 476-8: italics
Author Response
Dear Reviewers,
Thank you for your interest in our manuscript. We would like to thank the reviewers for taking the time to read and review the manuscript in so much depth. We are grateful for the remarks and helpful feedback which will certainly contribute to the quality of our manuscript. We have carefully addressed the comments and suggestions of the reviewers and revised the manuscript accordingly. Due to the scope we chose, the manuscript is not suited for some in depth explanations and discussions. Therefore, where possible we have made adjustments in the manuscript, and in other cases merely clarified in the rebuttal letter.
Below is out reply to each comment and description of the changes we made.
Reviewer 1:
> I have just one general comment to the results section. It is an impressive amount of data that was extracted and summarized by the authors. From the description of the search criteria it seems, that all the cases were confirmed by molecular biology methods. I am convinced this is another important piece of information from the clinical and also research point of view: sample type, timing of sampling, extraction and detection method used. Would it be possible to include this section to each of the pathogens?
>> We would like to thank the reviewer for this suggestion and acknowledge that for the identification of patients, it is crucial to perform the proper (laboratory) diagnostics at the right time and using the optimal methods. However the scope of our review is to provide insight into the evidence of pathogenicity of potential tick-borne pathogens, other than Lyme disease or TBE. Our search would be able to provide an answer to the optimal diagnostic approach, nevertheless to put this topic justice you would need to write a separate review. Therefore we decided to maintain the focus of our manuscript, But have added the molecular biology methods used for diagnosing the cases to section 2.1 “Literature Search”, as confirmation of infection was needed for the inclusion of a paper. (lines 131-133, 139-141)
> Line 76: Anaplasma strains – Anaplasma phagocytphilum strains?
>> We changed this to “A. phagocytophilum strains”. (line 77)
> Line 87 (and elsewhere): B. burgdorferi sl => B. burgdorferi s. l.
>> We corrected this throughout the manuscript.
> Line 105: Rickettsia’s => Rickettsia?
>> Corrected to Rickettsiae as it refers to both R. helvetica and R. monacensis. (line 106)
> Figure 1 – low quality (at least in the pdf generated for peer review).
>> Thank you for raising the point. The figure has been replaced with a different format. In case it is not sufficient the original figure can be provided.
> Line 138: an erythematous skin manifestations => erythematous skin manifestations
>> This has been adjusted. (line 145)
> Line 148: “…and exposure to ticks was reported in an additional 38% of cases“ => „…and exposure to ticks was reported in additional 38% of cases“ = so in total exposure to ticks was reported in 34 + 38 = 72% of cases?
>> The sentence has been edited and now reads: “Eleven (34%) patients recalled a tick-bite and twelve (38%) of the cases reported exposure to ticks”. (lines 154-155)
> Line 156: „infect human and host reservoir neutrophils“ => infect human and reservoir host or animal host neutrophils?
>> The word “animal” has been added. (line 162)
> Line 161: existing infection? = . pre-existing infection, co-infection?
>> This has been corrected. (line 167)
> Line 166: There is a report on transmission of canine strains of A. phagocytophilum to dogs by infected I. ricinus: https://parasitesandvectors.biomedcentral.com/articles/10.1186/s13071-019-3396-9 ; probably an European strain of A. phagocytophilum? + line 374,375
>> Thank you for this suggestion. The valuable evidence provided in the publication has been incorporated into the manuscript. (lines 172-173, tables 2,3)
> Line 235: Three European cases with B. miyamotoi suspected disease => with suspected B. miyamotoi disease (BMD)
>> This has been adjusted. (line 242)
> Line 289: N. mikurensis has been widely isolated from I. ricinus => N. mikurensis has been widely detected in I. ricinus?
>> This has been corrected to “molecularly detected in ”. (line 296)
> Line 296: R. helvetica => R. helvetica
>> This has been altered. (line 303)
Discussion
> Line 383: in cases from endemic regions with high fever => in cases from endemic regions and presenting with high fever
>> This has been adjusted. (line 390)
> Line 410-411: has not been cultivated from an I. ricinus acquired individual => From an individual infected by I. ricinus tick bite
>> This has been corrected to: “has not been cultivated from an individual infected by an I. ricinus acquired tick-bite”. (lines 417-418)
> Line 434: to elucidated on the specific pathology => to elucidate the specific pathology
>> This has been corrected. (line 441)
> Line 437: serologically discarded => rejected based on serology
>> This has been adjusted. (lines 444)
> Line 438: on an early disease stage => in an early disease stage
>> This has been corrected. (line 445)
> Line 442: „This however should not be mistaken with disease 442 causation which has not been demonstrated“ – I believe I understand what the authors try to say, but the sentence could be formulated better to improve understanding.
>> We understand and have edited the sentence and now reads: “However, proof found of a Rickettsia infection does not imply or translate to symptomatic disease. Therefore the evidence should not be mistaken for disease causation, which has not been demonstrated”. (lines 449-451)
> Line 467: bioinformatical analysis => bioinformatic analyses
>> This has been altered. (line 475)
Materials and Methods
Data extraction:
> Could you define the terms “tick bite” and “exposure to ticks” you use? E.g. the time frame of a tick bite? Does exposure mean a historical tick bite or a stay in tick habitat?
>> Thank you for asking the question. The following definition was added to section 4.3 Data extraction: “History of a tick-bite was recorded whenever a patient recalled a tick-bite on presentation. Exposure to ticks was recorded whenever a patient worked in a high risk occupation e.g. forest ranger, had engaged in activities considered to be high risk for a tick-bite such as hiking or had been exposed to animals.” (lines 496-499)
> Line 476-8: italics
>> This has been corrected. (lines 484-486)
Reviewer 2:
> Keywords, line 28: Ixodes ricinus in italic
>> This has been altered. (line 28)
- Introduction
> Line 63: the authors should be consistent and use A. phagocytophilum or Anaplasma phagocytophilum at the beginning of a sentence. It seems that the authors prefer to use the abbreviated form so please change all the sentences accordingly (such as in lines 367 and 374 that is Anaplasma phagocytophilum).
>> This has been adjusted throughout the manuscript.
> Line 64: United States – did you mean United States of America (USA)
>> This has been corrected throughout the manuscript.
> Line 71: ALT and AST should be first without abbreviation.
>> This has been corrected to: ”e.g. Alanine aminotransferase (ALT) or Aspartate transaminase (AST)”. (line 72)
> Line 72 and line 74: US or USA? (to be consistent with USA in the discussion).
>> This has been corrected to ”USA” throughout the manuscript.
- Results
> Anaplasma Phagocytophilum (in italic)
>> This has been corrected. (line 142)
> Line 250: BMD – first without abbreviation
>> This has been corrected. (line 242)
> 2.2.4. Neoehrlichia mikurensis (in italic)
>> This has been corrected. (line 269)
> Line 296: R. helvetica (in italic)
>> This has been corrected. (line 303)
> Table 1: I. ricinus (in italic), row 5 and column 7
>> This has been corrected.
- Materials and Methods:
> Lines 476-478: names in italic.
>> This has been corrected. (line 484-486)

Reviewer 2 Report
Review Report
Manuscript ID: pathogens-732391
Type of manuscript: Review
Title: Evaluation of disease causality of rare Ixodes ricinus-borne
infections in Europe
Authors: Tal Azagi, Dieuwertje Hoornstra, Kristin Kremer, Joppe W.R.
Hovius, Hein Sprong
Special Issue: New Frontiers in Tick Research
Brief summary:
The aim of this work was to gather evidence of human disease causality in Europe for Anaplasma phagocytophilum, Babesia divergens, Babesia microti, Babesia venatorum, Borrelia miyamotoi, Neoehrlichia mikurensis, Rickettsia helvetica and Rickettsia monacensis, microorganisms transmitted to humans through the bite of Ixodes ricinus tick. Based on literature review and case reports from 2008-2018 the authors found a strong evidence for disease causality in Europe by A. phagocytophilum and B. divergens but not for the remaining microorganisms. The authors also identified the required steps to get a comprehensive evidence for disease causality of the microorganisms in focus.
Broad comments:
In Europe, I. ricinus is the primary vector of important diseases affecting humans and animals. Diseases such as Lyme Borreliosis and Tick borne encephalitis are on the rise, leading to fear in our society and to media attention. Another concern that our society is facing is the potential disease caused by other microorganisms transmitted by I. ricinus: Anaplasma phagocytophilum, Babesia divergens, Babesia microti, Babesia venatorum, Borrelia miyamotoi, Neoehrlichia mikurensis, Rickettsia helvetica and Rickettsia monacensis. However, currently there are some knowledge gaps on the evidence of pathogenicity of these microorganisms to humans being this a timely work of vital importance to clarify what we really do know (is there a robust evidence that these microorganisms are pathogenic?) and identify directions of future research. The message of this review is applied not only to the research community but also to all citizens concerned with this topic.
The present review is extremely well written, clear and easy to read and has a good literature search.
Specific comments:
Minor changes (very minor):
Keywords, line 28: Ixodes ricinus in italic
- Introduction
Line 63: the authors should be consistent and use A. phagocytophilum or Anaplasma phagocytophilum at the beginning of a sentence. It seems that the authors prefer to use the abbreviated form so please change all the sentences accordingly (such as in lines 367 and 374 that is Anaplasma phagocytophilum).
Line 64: United States – did you mean United States of America (USA)
Line 71: ALT and AST should be first without abbreviation
Line 72 and line 74: US or USA? (to be consistent with USA in the discussion).
- Results
- Anaplasma Phagocytophilum (in italic)
Line 250: BMD – first without abbreviation
2.2.4. Neoehrlichia mikurensis (in italic)
Line 296: R. helvetica (in italic)
Table 1: I. ricinus (in italic), row 5 and column 7
Materials and Methods:
4.2. Selection Criteria
Lines 476-478: names in italic.
Author Response

(The authors gave the same response as above.)
